# Pro-Angiogenic Effects of Canine Platelet-Rich Plasma: In Vitro and In Vivo Evidence

**DOI:** 10.3390/ani15152260

**Published:** 2025-08-01

**Authors:** Seong-Won An, Young-Sam Kwon

**Affiliations:** Department of Veterinary Surgery, College of Veterinary Medicine, Kyungpook National University, Daegu 41566, Republic of Korea; sheep0308@knu.ac.kr

**Keywords:** angiogenesis, dog, platelet-rich plasma (PRP), rabbit corneal micropocket assay

## Abstract

Angiogenesis is a key component of the wound healing process. This study evaluated the angiogenic effects of canine platelet-rich plasma (cPRP) using in vitro and in vivo models. In vitro assays demonstrated that 20% cPRP significantly increased proliferation, migration and tube formation of human umbilical vein endothelial cells. In vivo angiogenesis was assessed using a rabbit corneal micropocket model. After implanting cPRP-containing pellets into the avascular cornea, cPRP induced neovascularization, with new blood vessels extending from the limbus toward the pellet. These findings suggest that cPRP may serve as a valuable therapeutic option for managing impaired wound healing in veterinary patients, including those with diabetes, aging-related conditions, or immune dysfunction.

## 1. Introduction

Angiogenesis, a critical aspect of neovascularization, plays an essential role in tissue repair and wound healing. This process is primarily driven by growth factors released from activated platelets, positioning platelet-rich plasma (PRP) as a promising modality in regenerative therapies [1,2,3]. Accordingly, investigating the angiogenic potential of canine-derived PRP (cPRP) is crucial to expanding its clinical utility in veterinary medicine.

PRP has gained attention as a regenerative agent due to its autologous origin and ability to deliver concentrated growth factors directly to sites of tissue injury [4,5,6,7]. In veterinary practice, cPRP has been used in the treatment of various musculoskeletal conditions and nonhealing wounds [8,9,10,11]. However, while the therapeutic effects of cPRP on tissue repair have been described, its direct influence on angiogenic pathways has not been rigorously studied.

Although numerous studies have explored the pro-angiogenic effects of human PRP, there remains a notable gap in the literature regarding species-specific responses—particularly in canines [5,12,13]. Most existing research has focused exclusively on human-derived PRP using human endothelial cell models, overlooking the biological variations that may influence therapeutic efficacy in animals [13,14]. This study addresses that gap by investigating the angiogenic effects of cPRP using in vitro human endothelial assays and an in vivo rabbit corneal micropocket model.

We hypothesized that cPRP enhances angiogenic activity in a concentration-dependent manner. To test this, we conducted proliferation, migration, and tube formation assays on HUVECs, followed by corneal implantation of cPRP-containing pellets in rabbits. This dual-model strategy allowed us to assess both cellular responses and visible vascular growth, establishing a comprehensive evaluation of cPRP’s angiogenic potential.

## 2. Materials and Methods

### 2.1. cPRP Preparation and Activation

Healthy Beagles were used for this study. Thirty-six milliliters of whole blood was collected aseptically from the jugular vein with a 50 mL syringe containing 4 mL of ACD-A solution (C.T.G. solution, Daehan New Pharm Co., Hwaseong, Republic of Korea). After transferring whole blood to 20 mL PRP kits (Pro PRP kit-20 cc, Goodmorning Bio Co., Incheon, Republic of Korea), cPRP was prepared according to the manufacturer’s protocol. cPRP was stored at −80 °C, thawed in a water bath at 37 °C for 30 min and centrifuged for 3 min at 2000 rpm at 37 °C before use. This freeze-thaw process was intended to achieve mechanical activation of PRP without the effect of additives [15,16].

### 2.2. Proliferation Assay

HUVECs were purchased from Corning Co. and cultured in endothelial cell growth medium (ECGM) (R&D Systems, Minneapolis, MN, USA). Cultures were incubated at 37 °C in a humidified atmosphere containing 5% CO_2_. Third passage HUVECs were seeded on 96-well culture plates (Corning Co., New York, NY, USA) with a density of 5.0 × 10^3^ cells per well. A proliferation assay was conducted using a Cell Counting Kit-8 (CCK-8) (Dojindo Molecular Technologies Inc., Kumamoto, Japan) according to the manufacturer’s protocol, and the assay was performed in triplicate. To assess the effects of PRP on HUVEC proliferation, HUVECs were treated with ECGM plus supplement containing 0%, 10%, and 20% cPRP and incubated for 6, 12, and 18 hours (h). The absorbance was measured at 450 nm using a microplate spectrophotometer (Epoch microplate spectrophotometer, Bioteck Instruments, Winooski, VT, USA).

### 2.3. Migration Assay

Migration was assessed using a 2-well Culture-Insert in a 35-mm µ-Dish (Ibidi Inc., Munich, Germany). To evaluate the migration ability of cPRP, 70 µL of HUVECs (5.0 × 10^5^ cells/mL) were seeded into each well. Once the cells reached 100% confluence, the 2-well Culture-Insert was gently removed using sterile tweezers, creating a 500 µm cell-free gap between the two confluent monolayers. Cell debris was carefully washed away with PBS, and the wells were then incubated for 24 h in ECGM plus supplement containing 0%, 10%, and 20% cPRP. To quantify the area of cell migration, phase contrast photographs were captured with a digital camera coupled to an inverted microscope (CKX41, Olympus Inc., Tokyo, Japan) at 6 and 12 h. Migration was measured using ImageJ software version 1.53t (National Institutes of Health, Bethesda, MD, USA). All experiments were repeated three times, and the result of the migration assay was converted to a percentage of the area where HUVECs migrated into the cell-free gap.

### 2.4. Tube Formation Assay

A tube formation assay was performed according to the manufacturer’s protocol. Briefly, Matrigel (Corning Co., New York, NY, USA) was thawed at 4 °C overnight. A volume of 300 µL of chilled Matrigel was dispensed into each well of a pre-cooled 24-well plate and incubated at 37 °C for 1 h to allow gelation. Subsequently, 300 µL of HUVECs (1.2 × 10^5^ cells) was added to each well, and the plates were incubated at 37 °C in a humidified atmosphere containing 5% CO_2_ for 12 h. Then, the medium was replaced with 300 µL of ECGM plus supplement containing 0%, 10%, and 20% cPRP. After 24 h of incubation, phase contrast photographs were captured with a digital camera coupled to an inverted microscope (CKX41, Olympus Inc., Tokyo, Japan). Tube numbers were counted at 40× magnification and averaged to obtain quantitative data. The assay was performed in triplicate.

### 2.5. Rabbit Corneal Micropocket Assay

This study was approved by the Ethics Committee of Kyungpook National University (Approval number: 2020-0120). Male New Zealand White rabbits (*n* = 8, 12 weeks of age, 2.99 ± 0.24 kg) were housed under constant temperature and lighting conditions (23 ± 2 °C, 12/12-h light/dark cycle) and fed a commercial rabbit pellet diet with tap water provided ad libitum. The corneal micropocket assay was conducted according to a previously described method [17,18]. Eight rabbits were randomly assigned to four groups, with each group consisting of four eyes from two rabbits: the control group, in which pellets without cPRP were inserted, and the 10%, 20%, and 40% cPRP-treated groups. Rabbits were anesthetized by intramuscular injection of ketamine hydrochloride (35 mg/kg) and xylazine (5 mg/kg). Before inserting a pellet into the cornea, local anesthesia was achieved by applying 0.5% proparacaine hydrochloride (Alcon Laboratories, Fort Worth TX, USA) on the corneal surface, and 0.3% ofloxacin eyedrops (Samil Pharmaceutical, Seoul, Republic of Korea) were administered 3 times per day for 14 days after surgery. Corneal angiogenesis was monitored and photographed at 16× magnification with a slit lamp stereomicroscope (SM-70N Slit lamp microscope, Takagi-Seiko Co., Nagano, Japan).

### 2.6. Statistical Analysis

All data are expressed as the mean ± SD. Three replicates were performed for each experiment. Statistical analyses were performed using GraphPad Prism 8 (GraphPad Software Inc., San Diego, CA, USA). Prior to analysis, data normality was evaluated using the Shapiro–Wilk test, and homogeneity of variance was assessed using Levene’s test. One-way ANOVA was used for group comparisons, followed by Tukey’s post hoc test when significance was detected. A *p*-value of less than 0.05 was considered statistically significant. *p* values are represented as * *p* < 0.05 and ** *p* < 0.01 to indicate statistically significant differences.

## 3. Results

### 3.1. Proliferation Assay

In the CCK-8 assay, the proliferation of HUVECs increased in a cPRP dose-dependent manner when the cells were treated with 10% and 20% cPRP. Compared with the control group, the cell proliferation of the 20% cPRP group markedly increased by 41.94%, 38.24%, and 40.54% at 6, 12, and at 18 h, respectively (*p* < 0.05 or *p* < 0.01). At 12 h, the 20% cPRP group showed a statistically significant increase of 34.29% compared to the 10% cPRP group (*p* < 0.05) (Figure 1).

### 3.2. Migration Assay

Cell migration was significantly higher in the 20% cPRP group than in the control or 10% cPRP group at 6 and 12 h (*p* < 0.05 or *p* < 0.01). Furthermore, 10% and 20% cPRP significantly increased the migration rate of HUVECs by 30.17% and 54.22% at 6 h, and by 61.14% and 86.31% at 12 h, respectively (Figure 2).

### 3.3. Tube Formation Assay

In the Matrigel tube formation assay, the quantity of tubes markedly increased in the cPRP groups compared with the control group. At 24 h, the number of endothelial tubes was 62.83 ± 4.45 in the control group, 74.00 ± 2.28 in the 10% cPRP group, and 81.17 ± 6.52 in the 20% cPRP group. Both cPRP groups showed a statistically significant increase compared to the control (*p* < 0.01), and the 20% cPRP group exhibited a significantly higher tube formation than the 10% cPRP group (*p* < 0.05) (Figure 3).

### 3.4. Rabbit Corneal Micropocket Assay

This assay was conducted to confirm whether cPRP can directly induce angiogenesis. Angiogenesis occurred in all eyes in which cPRP-containing pellets were inserted, and the degree was proportional to the cPRP concentration (Figure 4). In the control group, no angiogenesis occurred during the experimental period. In the 10% cPRP-treated group, mild angiogenesis from the upper limbus was confirmed on day 14. Angiogenesis was confirmed on days 10 and 14 in the 20% cPRP-treated group. In the 40% cPRP-treated group, sprouting vessels from the upper limbus were observed from days 1 to 3, and there were many more vessels than in the 10% and 20% cPRP-treated groups. The corneal micropocket assay was initially performed with control, 10%, and 20% PRP groups. However, clear angiogenesis in the corneal stroma was not observed in the 20% PRP group. Therefore, a 40% cPRP group was added, which provided clear and definitive evidence of the angiogenic effect of cPRP in the cornea.

## 4. Discussion

This study provides novel evidence that canine platelet-rich plasma (cPRP) directly promotes angiogenesis, as demonstrated by enhanced endothelial proliferation, migration, and tube formation in vitro, along with dose-dependent neovascularization in vivo. While PRP has long been recognized for its regenerative properties in both human and veterinary contexts [4,5,12], few studies have systematically investigated its pro-angiogenic potential in canines using both cellular and animal models.

Our findings are in line with previous research on human PRP, which showed enhanced angiogenic responses in endothelial cells [13,19]. Similar to our results, human PRP has been reported to induce endothelial tube formation at concentrations ranging from 10% to 50% [19,20]. However, our data suggest that canine PRP requires higher concentrations to elicit comparable in vivo effects, as robust neovascularization in the rabbit corneal micropocket assay was only evident with 40% cPRP. This discrepancy may be attributed to species-specific differences in platelet content or growth factor profiles [21], reinforcing the need for species-matched validation before clinical application.

The in vivo component of this study employed a well-established corneal micropocket assay, which offers a high-resolution model for observing angiogenesis in an avascular environment [18]. We observed that cPRP induced early sprouting from the limbus as early as day 1, with the extent and intensity of neovascularization correlating with cPRP concentration. This aligns with prior studies using VEGF or PDGF pellets in the same model [17,22], though the growth factor concentrations in purified systems often exceed those found in native PRP. Notably, our results showed that even without external activators such as thrombin or CaCl_2_, freeze-thawed cPRP maintained sufficient angiogenic activity [15,16].

Nevertheless, several limitations must be acknowledged. First, the sample size (n = 8) may restrict statistical power and limit broader generalizability. Second, angiogenesis was monitored only for 14 days, precluding assessment of long-term vessel maturation or remodeling. Third, while HUVECs are a standard endothelial model, they may not fully replicate canine-specific responses. Furthermore, we did not quantify specific angiogenic growth factors such as VEGF, PDGF, or FGF in our cPRP samples—information that could help correlate biological composition with functional outcomes [23,24].

Clinically, cPRP shows considerable promise for treating chronic wounds or ischemic tissues in veterinary patients, particularly geriatric or diabetic dogs, or those with impaired immune responses [8,9,10]. Potential routes of administration include subcutaneous or intralesional injection near wound sites, intra-articular delivery in degenerative joint disease [9], or use as a topical gel or biomaterial coating during surgery [7,11]. However, standardized protocols regarding dosage, frequency, and activation methods remain to be defined.

Future research should involve canine-specific wound or ischemia models to verify efficacy in autologous contexts. Additionally, long-term studies should evaluate vessel stability, integration into host vasculature, and overall functional outcomes. Comparative analyses of PRP derived from different species, breeds, or individuals may also reveal variability that influences therapeutic success [21].

In summary, this study is the first to provide integrated in vitro and in vivo evidence that cPRP can induce angiogenesis in a concentration-dependent manner. These findings establish a foundation for future clinical trials and underscore the therapeutic potential of cPRP as a regenerative tool in veterinary medicine.

## 5. Conclusions

Conclusively, we showed that cPRP exhibits angiogenic effects using in vitro and in vivo models. In the in vitro study, cPRP substantially enhanced HUVEC proliferation and migration as well as vascular endothelial tube formation. In addition, corneal angiogenesis was stimulated by cPRP pellet implantation using an in vivo corneal micropocket assay. Therefore, cPRP, which can be prepared autologously, may be very useful for the treatment of delayed wound healing in diabetic animals, those with immune-mediated disease, and geriatric animals. Future studies should clarify strategies for clinical application in animals with nonhealing wounds.

## Figures and Tables

**Figure 1 animals-15-02260-f001:**
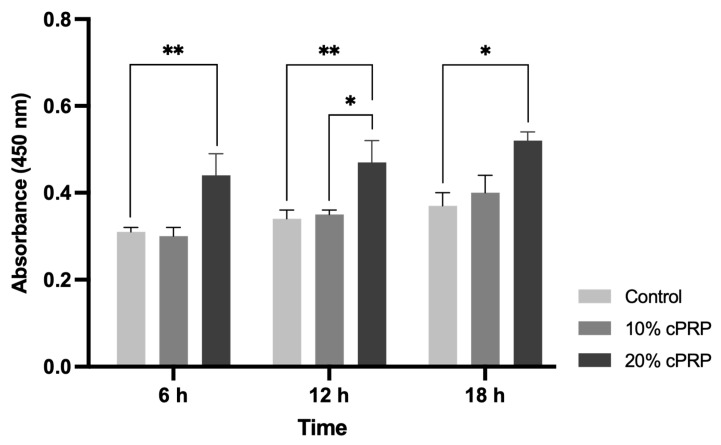
Effect of cPRP on HUVEC proliferation. Data are expressed as the mean ± SD. * *p* < 0.05, ** *p* < 0.01.

**Figure 2 animals-15-02260-f002:**
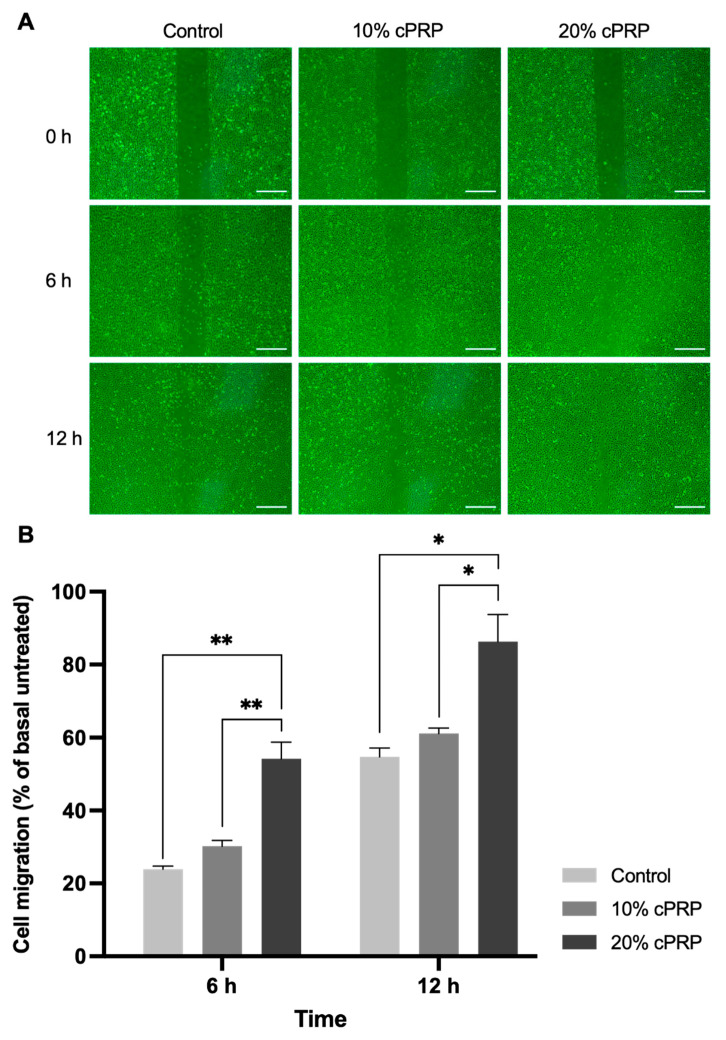
Results of the migration assay of HUVECs. Microscopic features of the migration assay (**A**). Photographs of the migration of HUVECs into the cell-free gap were taken at 6 and 12 h after PRP treatment using a digital camera coupled to an inverted microscope (×40). Results of the migration assay of HUVECs (**B**). Data are expressed as the mean ± SD. * *p* < 0.05, ** *p* < 0.01. Scale bars: 500 µm.

**Figure 3 animals-15-02260-f003:**
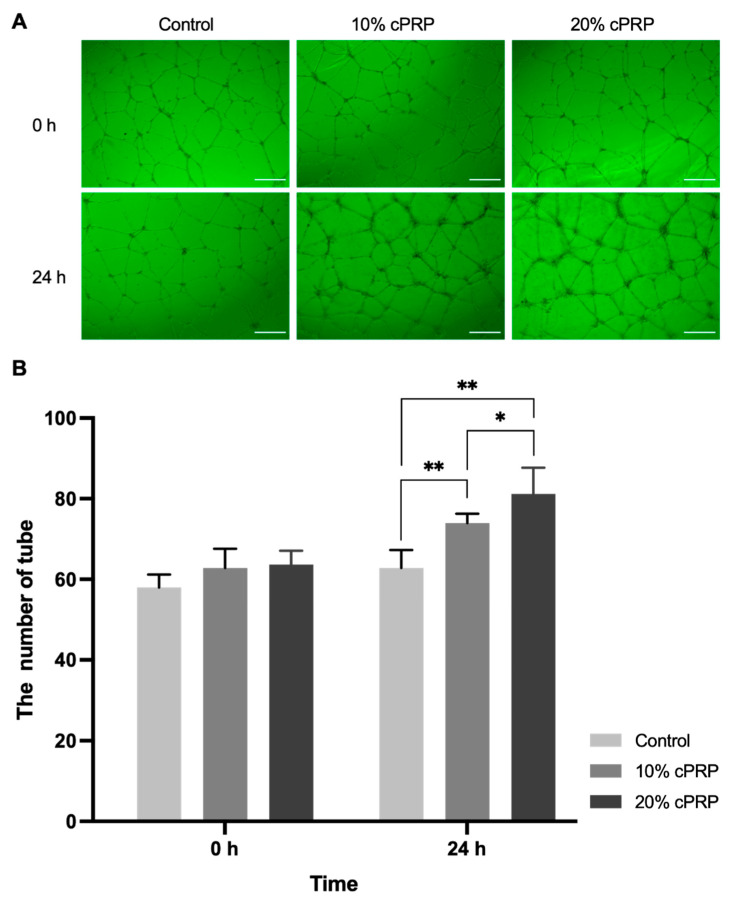
Microscopic features of the tube formation assay (**A**). Results of the tube formation assay (**B**). Data are expressed as the mean ± SD. * *p* < 0.05, ** *p* < 0.01. Scale bars: 500 µm.

**Figure 4 animals-15-02260-f004:**
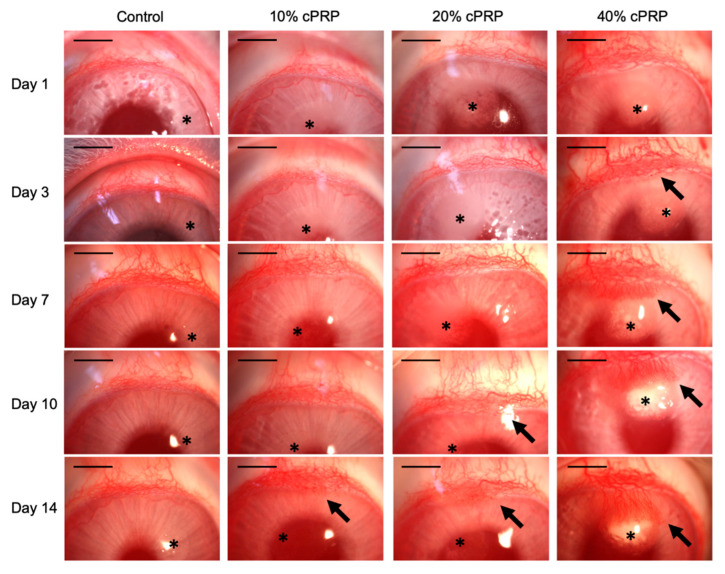
Slit lamp photographs of the rabbit corneal micropocket assay (×16). Photographs of eyes were collected on days 1, 3, 7, 10, and 14 after pellet insertion. Black arrow, grown capillary; *, pellet. Scale bars: 5 mm.

## Data Availability

The original contributions presented in this study are included in the article. Further inquiries can be directed to the corresponding author(s).

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
