# Peer review of "Pro-Angiogenic Effects of Canine Platelet-Rich Plasma: In Vitro and In Vivo Evidence"

_animals, 2025, doi:10.3390/ani15152260_

Round 1

Reviewer 1 Report

Comments and Suggestions for Authors

Dear authors,

The work you present is an interesting contribution. With a view to improving the final version, here are my suggestions.

Keywords should not repeat words in the title: for example, dog instead of canine.

Simple summary

Line 9 - and tissue regeneration

Line 14 - such as the rabbit cornea.

Introduction

Line 34 - Vasculogenesis is the de novo formation of new blood vessels from stem cells during early embryonic development. Angiogenesis is the growth of new blood vessels from pre-existing vessels, primarily capillaries. 

In our opinion and in the context of the paragraph, it makes sense to write the notion of arteriogenesis.

Arteriogenesis, on the other hand, involves the enlargement of pre-existing arterioles to form larger arteries. 

Results

Lines 116, 127, 136 - please write statistical p in italics

Line 132 - Figure 3 is not available in the present version of the work.

The use of varying concentrations of PRP depending on the trials or the reporting of results for only a few concentrations must be duly justified (0%-10%-20% and 10%-20%-40%)

Figure 4 - In relation to figure 4 and with the exception of the 40% concentration, the confluence of the new vessels with the pellet does not seem evident in the other photos relating to the other concentrations. In this sense better photographic evidence must be provided.

Discussion

Line 171 - "We used the freeze-thaw method, which is a mechanical activation method, to exclude the effect of the additives". This sentence should be inserted in the materials and methods section.

Author Response

Reviewer 1

Comments 1: Keywords should not repeat words in the title: for example, dog instead of canine.

Response 1: Thank you for pointing this out. We agree with this comment. Therefore, we revised the keyword from "canine" to "dog". This change can be found – line 27.

Comments 2: Line 9 - and tissue regeneration

Response 2: Thank you for pointing out the typos. Following a suggestion from another reviewer, I have revised and rewritten the simple summary.

Comments 3: Line 14 - such as the rabbit cornea

Response 3: Thank you for pointing out the typos. Following a suggestion from another reviewer, I have revised and rewritten the simple summary.

Comments 4: Line 34 - In our opinion and in the context of the paragraph, it makes sense to write the notion of arteriogenesis. Arteriogenesis, on the other hand, involves the enlargement of pre-existing arterioles to form larger arteries. 

Response 4: Thank you for pointing this out. As you suggested, the following has been inserted into the manuscript; “Arteriogenesis, on the other hand, involves the enlargement of pre-existing arterioles to form larger arteries”- lines 36-37.

Comments 5: Lines 116, 127, 136 - please write statistical p in italics

Response 5: Thank you for pointing this out. As you requested, we changed “P” to “P”.

Comments 6: Line 132 - Figure 3 is not available in the present version of the work.

Response 6: As you point this out, we added “Figure 3”. Thank you.

Comments 7: The use of varying concentrations of PRP depending on the trials or the reporting of results for only a few concentrations must be duly justified (0%-10%-20% and 10%-20%-40%)

Response 7: Thank you for your evalution. As you pointed out, we added the following explanation: “The corneal micropocket assay was initially performed with control, 10%, and 20% PRP groups. However, clear angiogenesis into the corneal stroma was not observed in the 20% PRP group. Therefore, a 40% PRP group was added, which provided clear and definitive evidence of the angiogenic effect of PRP in the cornea. ” – lines 152-156

Comments 8: Figure 4 - In relation to figure 4 and with the exception of the 40% concentration, the confluence of the new vessels with the pellet does not seem evident in the other photos relating to the other concentrations. In this sense better photographic evidence must be provided.

Response 8: We appreciate the reviewer’s valuable comment. We prepared pellets using 12% polyhydroxyethylmethacrylate (pHEMA) in alcohol and implanted them into the corneal stroma. When PRP was added at varying concentrations, the pellets appeared clearer in the cornea at lower concentrations, while higher concentrations resulted in visibly more opaque pellets.

In the 0–20% PRP groups, we photographed the same cornea over a 14-day period to monitor the progression of neovascularization. Although angiogenesis was observed in the initial corneal micropocket assay (control, 10% PRP, and 20% PRP groups), we conducted an additional experiment with a 40% PRP group to provide more definitive evidence. As a result, we were able to demonstrate that 40% PRP clearly induced angiogenesis in the normally avascular corneal tissue.

We hope the reviewer will kindly consider these points in understanding our approach.

Comments 9: Line 171 - "We used the freeze-thaw method, which is a mechanical activation method, to exclude the effect of the additives". This sentence should be inserted in the materials and methods section.

Response 9: We appreciate your insightful comment. In accordance with your suggestion, the content has been added to the Materials and Methods section. “This freeze-thaw process was intended to achieve mechanical activation of PRP without the effect of additives.”- lines 58-59

Reviewer 2 Report

Comments and Suggestions for Authors

Title

Please, change the title of the manuscript. Now it sounds like part of a text. May be something like one of this one:

Pro-Angiogenic Effects of Canine Platelet-Rich Plasma: In Vitro and In Vivo Evidence"  

Or think of other option.

Simple summary

Please, rewrite ss section,  

Abstract 

rewrite abstract

Introduction

Rewrite Introduction, It’s not clear right now what did you do and what was the purpose of research?

Add list of abbreviations

methods 

Please provide ELISA or proteomic profiling data to demonstrate the concentration of key angiogenic factors in your cPRP batches.

describe negative and positive controls 

A positive control is critical to demonstrate the angiogenic efficacy of cPRP.

Expand on the activation protocol and include data (e.g., platelet count pre- and post-processing, activation markers) or cite standardized protocols.

The number of replicates (n=2–3 in in vitro, and only n=2 per group in vivo) is too low to draw statistically sound conclusions.

Results

rewrite this section

Figures 2 and 4

I don’t see any difference. Please, show some details on figures, add scale bars. Figure 2 - what did you mean? Mark the difference, right now I could see only green color. Figures lack scale bars, proper labels, and statistical annotations (e.g., sample size, exact p-values). Imrove visual clarity. Add details

Author Response

Reviewer 2

Comments 1: Title; Please, change the title of the manuscript. Now it sounds like part of a text. May be something like one of this one: Pro-Angiogenic Effects of Canine Platelet-Rich Plasma: In Vitro and In Vivo Evidence" Or think of other option.

Response 1: We appreciate your insightful comment. In accordance with your suggestion, we changed the title as “Pro-Angiogenic Effects of Canine Platelet-rich Plasma: In Vitro and In Vivo Evidence”

Comments 2: Simple summary. Please, rewrite ss section,  

Response 1: We appreciate your insightful comment. In accordance with your suggestion, we rewrite Simple Summary.

Comments 3: Abstract.Rewrite abstract

Response 3: We appreciate your insightful comment. In accordance with your suggestion, We partially revised the abstract.

Comments 4: Introduction. Rewrite Introduction, It’s not clear right now what did you do and what was the purpose of research? Add list of abbreviations methods 

Response 4: We appreciate your insightful comment. In accordance with your suggestion, we rewrite Introduction.

Comments 5: Please provide ELISA or proteomic profiling data to demonstrate the concentration of key angiogenic factors in your cPRP batches.

Response 5: Thank you for your comment. We are currently conducting a quantitative analysis of key growth factors in cPRP, including PDGF, VEGF, FGF2, and TGF-β1. Although the data are not yet available for presentation, preliminary ELISA results indicate the presence of all four factors. We plan to publish these findings in a future study.

Comments 6: describe negative and positive controls. A positive control is critical to demonstrate the angiogenic efficacy of cPRP.

Response 6: Thank you for your insightful comments. This study was designed as a preliminary investigation to evaluate the potential angiogenic effects of cPRP using both in vitro and in vivo models. Regarding your comment on the positive control, we would like to clarify that the controls used in both in vitro and in vivo experiments served as the reference for this evaluation. We sincerely appreciate your suggestion and will make sure to incorporate this important consideration in our future PRP studies.

Comments 7: Expand on the activation protocol and include data (e.g., platelet count pre- and post-processing, activation markers) or cite standardized protocols.

Response 7: We sincerely appreciate your thoughtful suggestion. In accordance with your request, we have now cited appropriate references related to the PRP activation method. Lines 59-60

Comments 8: The number of replicates (n=2–3 in in vitro, and only n=2 per group in vivo) is too low to draw statistically sound conclusions.

Response 8: Thank you for your valuable comment. The in vitro experiments were conducted with n=3, and the in vivo experiments were statistically analyzed using 4 eyes per group. This information has been clearly stated in the revised manuscript.

Comments 9: Results. rewrite this section

Response 9: Thank you for your valuable comment. As suggested, we have revised the Results section accordingly.

Comments 10: Figures 2 and 4

I don’t see any difference. Please, show some details on figures, add scale bars. Figure 2 - what did you mean? Mark the difference, right now I could see only green color. Figures lack scale bars, proper labels, and statistical annotations (e.g., sample size, exact p-values). Imrove visual clarity. Add details

Response 10: Thank you for your precise observation. The resolution of Figures 2 and 4 was reduced during the journal’s editing process. We have now replaced them with high-resolution images, and included the magnification details and significance indicators in both the figures and their legends.

Reviewer 3 Report

Comments and Suggestions for Authors

Dear Authors,

It was a pleasure to read your manuscript; however, there are several critical concerns that need to be addressed.

Introduction:

Line 32: The author mentioned that the Neovascularization includes three pathways: vasculogenesis, angiogenesis, and arteriogenesis.  Indeed, this sentence should be double-checked since neovascularization mainly refers to the creation of new vessels in two pathways: vasculogenesis and angiogenesis, but Arteriogenesis refers to the remodeling of an existing artery to increase its luminal diameter.

Line 45: ‘‘there‘‘ is misplaced in the sentence

Methods:

Line 58: In the method section, it is mentioned that the HUVECs were cultured on ECGM; please also add if the culture medium was supplemented with other additives like growth factors, serum albumin, Vitamins, antibiotic-antimycotics, or ….

Moreover, in the same section (2.2. Proliferation assay), the authors assessed the effects of PRP on HUVEC proliferation with different treatments of ECGM+cPRP and at different time intervals. The statistical analysis was also performed to compare the groups. The question is each groups contained how many cultures? This info should be given in the text.

The same issue occurs in the sections Migration and Tube formation assays. No information was given regarding the number of cultures that were compared with each other.

Line 90: Eight rabbits were randomly divided into four groups for the corneal micropocket assay. Were there two rabbits in each group? Please write the exact number of cases per group.

Statistical analysis: Why did you perform Dunnett’s test post hoc correction just to compare the treatment groups with the control one? Instead, you could compare all the groups, even the test groups of different times, with each other, using repeated-measure ANOVA.

Result:

3.1: It seems that there was no significant difference between the 10% cPRP treatment group and the control group. Specifically, after 6 hours, the graph shows less proliferation in the 10% cPRP treatment group compared to the control group. Please include this information in the text.

Did you also compare the 20% and 10% cPRP treatment groups with each other? According to the graph, a significant difference may be apparent between these two groups as well.

Since you have already provided detailed information about the differences between groups in the main text, it is not necessary to repeat it in the figure legend.

Figure 3 is missing! It is mentioned in the text (Line 132), but where is it?

3.4: You should indicate in the method section that the result of the Rabbit corneal micropocket assay will be reported descriptively due to the low number of sample size.

Fig 4. The titles of the image columns are better given at the top of each column instead of at their bottom. Moreover, as I mentioned in Fig. 1, please avoid repetition of all the info that was given in the text.

Additional general points:

  1. The formal abbreviation for hours in scientific writing ish, not hr.
  2. In the graph’s color legend, “PRP” is written instead of “cPRP.” Please correct this for consistency.
  3. For greater clarity, I also suggest using both lines and asterisks to indicate statistically significant differences between the selected groups in all the graphs.

Author Response

Reviewer 3

Introduction

Comment:
Line 32: The author mentioned that the Neovascularization includes three pathways: vasculogenesis, angiogenesis, and arteriogenesis.  Indeed, this sentence should be double-checked since neovascularization mainly refers to the creation of new vessels in two pathways: vasculogenesis and angiogenesis, but Arteriogenesis refers to the remodeling of an existing artery to increase its luminal diameter.

Response: Thank you for your insightful observation. As correctly pointed out, we have revised the sentence to reflect this distinction. Lines 36-37

Comment:
Line 45: ‘‘there‘‘ is misplaced in the sentence.

Response: We appreciate your careful reading. The word “there” has been removed from the sentence.

Methods

Comment:
Line 58: In the method section, it is mentioned that the HUVECs were cultured on ECGM; please also add if the culture medium was supplemented with other additives like growth factors, serum albumin, Vitamins, antibiotic-antimycotics, or ….

Response: Thank you for your comment. We have updated the Methods section to specify that ECGM used in the HUVEC culture was a complete medium plus supplement in the text.

Comment:
Moreover, in the same section (2.2. Proliferation assay), the authors assessed the effects of PRP on HUVEC proliferation with different treatments of ECGM+cPRP and at different time intervals. The statistical analysis was also performed to compare the groups. The question is each groups contained how many cultures? This info should be given in the text.

The same issue occurs in the sections Migration and Tube formation assays. No information was given regarding the number of cultures that were compared with each other.

Response: We appreciate this important point. All in vitro experiments (proliferation, migration, and tube formation assays) were independently repeated three times. This information has been added to the relevant sections of the Methods. We also have reanalyzed the statistical data for all groups and added the revised results to the Results section.

Comment:
Line 90: Eight rabbits were randomly divided into four groups for the corneal micropocket assay. Were there two rabbits in each group? Please write the exact number of cases per group.

Response:
Thank you for your observation. The sentence was revised as follows: “Eight rabbits were randomly divided into the following four groups, with four eyes in each group.”(line 94)

Comment:
Statistical analysis: Why did you perform Dunnett’s test post hoc correction just to compare the treatment groups with the control one? Instead, you could compare all the groups, even the test groups of different times, with each other, using repeated-measure ANOVA.

Response:
We appreciate the reviewer’s suggestion. We have reanalyzed the statistical data for all groups and added the revised results to the Results section and figures.

Results

Comment:
3.1: It seems that there was no significant difference between the 10% cPRP treatment group and the control group. Specifically, after 6 hours, the graph shows less proliferation in the 10% cPRP treatment group compared to the control group. Please include this information in the text.

Response:
Thank you for pointing this out. At 6 hours, the absorbance values were 0.31 ± 0.01 for the control group and 0.30 ± 0.02 for the 10% cPRP group, indicating no meaningful difference. Given the concise format of this communication, only statistically significant findings were reported. In addition, statistical analyses were conducted across all groups, and a newly identified statistically significant difference between the 10% and 20% groups at 12 hours was added.

 <Absorbance of proliferation of HUVECS at 6h, 12h, and 18 h>

Comment:
Did you also compare the 20% and 10% cPRP treatment groups with each other? According to the graph, a significant difference may be apparent between these two groups as well.

Response:
As you mentioned, pairwise comparisons between the 10% and 20% cPRP groups were conducted, and a statistically significant difference in proliferation, migraiont and tube for was observed. This information has now been included in the revised Results section.

Comment:
Since you have already provided detailed information about the differences between groups in the main text, it is not necessary to repeat it in the figure legend.

Response:
We agree and have removed redundant text from the figure legends, keeping only essential information not already presented in the main text.

Comment:
Figure 3 is missing.

Response:
We apologize for the oversight. Figure 3 has now been included in the revised manuscript.

Comment:
3.4: You should indicate in the method section that the result of the Rabbit corneal micropocket assay will be reported descriptively due to the low number of sample size.

Response:
Thank you for this suggestion. As another reviewer also requested clarification on the sample size used in the corneal micropocket assay, we have included additional details regarding the number of samples in the Methods section as follows; “Eight rabbits were randomly divided into the following four groups, with four eyes in each group.“

Comment:
Fig 4. The titles of the image columns are better given at the top of each column instead of at their bottom. Moreover, as I mentioned in Fig. 1, please avoid repetition of all the info that was given in the text.

Response:
Thank you for your comments. As you suggested, we have revised Figure 4 by repositioning the column headings to the top and streamlining the legend to remove repetition.

General Comments

  1. Comment:
    The formal abbreviation for hours in scientific writing ish, not hr.

Response:
We have revised all instances of "hr" to the standardized abbreviation "h" throughout the manuscript.

  1. Comment:
    In the graph’s color legend, “PRP” is written instead of “cPRP.” Please correct this for consistency.

Response:
Thank you. We have corrected all instances where “PRP” was used in the legends, ensuring consistency with “cPRP.”

  1. Comment:
    For greater clarity, I also suggest using both lines and asterisks to indicate statistically significant differences between the selected groups in all the graphs.

Response:
We appreciate the suggestion. In all figures, we have now added lines alongside asterisks to better visualize significant differences between groups.

Round 2

Reviewer 1 Report

Comments and Suggestions for Authors

Dear authors,

All my observations were answered satisfactorily.

Kind regards

Author Response

Reviewer 1

Comments 1: Keywords should not repeat words in the title: for example, dog instead of canine.

Response 1: Thank you for pointing this out. We agree with this comment. Therefore, we revised the keyword from "canine" to "dog". This change can be found – line 27.

Comments 2: Line 9 - and tissue regeneration

Response 2: Thank you for pointing out the typos. Following a suggestion from another reviewer, I have revised and rewritten the simple summary.

Comments 3: Line 14 - such as the rabbit cornea

Response 3: Thank you for pointing out the typos. Following a suggestion from another reviewer, I have revised and rewritten the simple summary.

Comments 4: Line 34 - In our opinion and in the context of the paragraph, it makes sense to write the notion of arteriogenesis. Arteriogenesis, on the other hand, involves the enlargement of pre-existing arterioles to form larger arteries. 

Response 4: Thank you for pointing this out. As you suggested, the following has been inserted into the manuscript; “Arteriogenesis, on the other hand, involves the enlargement of pre-existing arterioles to form larger arteries”- lines 36-37.

Comments 5: Lines 116, 127, 136 - please write statistical p in italics

Response 5: Thank you for pointing this out. As you requested, we changed “P” to “P”.

Comments 6: Line 132 - Figure 3 is not available in the present version of the work.

Response 6: As you point this out, we added “Figure 3”. Thank you.

Comments 7: The use of varying concentrations of PRP depending on the trials or the reporting of results for only a few concentrations must be duly justified (0%-10%-20% and 10%-20%-40%)

Response 7: Thank you for your evalution. As you pointed out, we added the following explanation: “The corneal micropocket assay was initially performed with control, 10%, and 20% PRP groups. However, clear angiogenesis into the corneal stroma was not observed in the 20% PRP group. Therefore, a 40% PRP group was added, which provided clear and definitive evidence of the angiogenic effect of PRP in the cornea. ” – lines 152-156

Comments 8: Figure 4 - In relation to figure 4 and with the exception of the 40% concentration, the confluence of the new vessels with the pellet does not seem evident in the other photos relating to the other concentrations. In this sense better photographic evidence must be provided.

Response 8: We appreciate the reviewer’s valuable comment. We prepared pellets using 12% polyhydroxyethylmethacrylate (pHEMA) in alcohol and implanted them into the corneal stroma. When PRP was added at varying concentrations, the pellets appeared clearer in the cornea at lower concentrations, while higher concentrations resulted in visibly more opaque pellets.

In the 0–20% PRP groups, we photographed the same cornea over a 14-day period to monitor the progression of neovascularization. Although angiogenesis was observed in the initial corneal micropocket assay (control, 10% PRP, and 20% PRP groups), we conducted an additional experiment with a 40% PRP group to provide more definitive evidence. As a result, we were able to demonstrate that 40% PRP clearly induced angiogenesis in the normally avascular corneal tissue.

In addition, we replaced figure with a high resolution photo.

We hope the reviewer will kindly consider these points in understanding our approach.

Comments 9: Line 171 - "We used the freeze-thaw method, which is a mechanical activation method, to exclude the effect of the additives". This sentence should be inserted in the materials and methods section.

Response 9: We appreciate your insightful comment. In accordance with your suggestion, the content has been added to the Materials and Methods section. “This freeze-thaw process was intended to achieve mechanical activation of PRP without the effect of additives.”- lines 58-59

Thank you for your thorough and enthusiastic review. The revised sections in the manuscript have been highlighted in red text.

Best regards,

Kwon

Reviewer 2 Report

Comments and Suggestions for Authors

Simple summary

Version 2 is a great improvement. It sounds like a good simple summary. However, it might be improved in some respects. Change the phrase “new blood vessels were observed” into a more direct phrase “ induced neovascularization”. Briefly state potential application. 

Abstract

Actually, I could see no major differences between the two versions. Rewrite abstract section, it lacks structure. State significance of the study,  add concluding sentences, highlight broader implications. The text repeats cPRP and angiogenesis three times, it’s unnecessary. Why PRP is important in veterinary medicine. State the novelty of the work clearly. Remove passive tone from text. write in an active tone. Rewrite abstract so that you follow the structure^ background-methods-results-conclusion. 

Introduction

In the revised version you added the difference between arteriogenesis and angiogenesis. 

Rewrite the introduction so that you emphasize the novelty of your research. Write that previous angiogenesis studies were focused on human PRP, not canine. 

The first paragraph is too broad and textbook like.

At the end of the introduction write about your hypothesis and overview study design. 

Methods

This section was improved. However, I have some comments:

Provide a detailed description for tube formation assay and  migration assay. Describe specific parameters for image analysis. How was quantification performed?

Replication is mentioned inconsistently in all experiments. I don’t understand whether different rabbits were used for each pellet type or  were eyes treated independently?

For statistical analysis add information about software used, normality test, variance homogeneity, how sample size were determined.

Results

This section improved in statistical representation, clarity, and better explanation.

Discussion

Rewrite discussions section. It has improved. However, I do have comments.

The discussion repeats information that already was explained in the introduction section. Interpret your actual findings. 

You mainly describe what you observed without deeper explanation. Please, provide comparative analysis of your results. 

Provide a critical analysis of limitations: small sample size (n=8 rabbits), lack of long-term angiogenesis data. No discussion. This weakens the results. Discuss clinical application and future directions. Discuss how cprp would be delivered into practice( injection or other?) .

What is the next step towards clinical translation?

Provide a critical comparative analysis of you results with previous studies.

Figures 

Still lack of consistent labeling, especially graphs

Add scale bars into migration and tube formation assays and microscopical images.

There is too much white space in some graphs. 

Check consistency of placing one or double asterisks on graphs.

Author Response

Author’s reply for Reviewer 2.

We would like to express our sincere gratitude to the reviewer for the thoughtful and thorough feedback. Please find our detailed responses below.

All changes made based on the reviewer’s suggestions have been marked in blue in the revised manuscript.

Reviewer 2

Comments and Suggestions for Authors

Simple summary

Version 2 is a great improvement. It sounds like a good simple summary. However, it might be improved in some respects. Change the phrase “new blood vessels were observed” into a more direct phrase “ induced neovascularization”. Briefly state potential application. 

Response: In accordance with the reviewer’s valuable suggestions, we have rewritten the Simple Summary to fully reflect the points raised.

Abstract

Actually, I could see no major differences between the two versions. Rewrite abstract section, it lacks structure. State significance of the study,  add concluding sentences, highlight broader implications. The text repeats cPRP and angiogenesis three times, it’s unnecessary. Why PRP is important in veterinary medicine. State the novelty of the work clearly. Remove passive tone from text. write in an active tone. Rewrite abstract so that you follow the structure^ background-methods-results-conclusion. 

Response: We appreciate the reviewer’s constructive comments and have rewritten the Abstract accordingly to reflect all suggestions provided.

Introduction

In the revised version you added the difference between arteriogenesis and angiogenesis. 

Rewrite the introduction so that you emphasize the novelty of your research. Write that previous angiogenesis studies were focused on human PRP, not canine. 

The first paragraph is too broad and textbook like.

At the end of the introduction write about your hypothesis and overview study design. 

Response: In appreciation of the reviewer’s suggestions, we have rewritten the Introduction to highlight the originality of our work, underscore the importance of cPRP, and incorporate the hypothesis and study outline.

Methods

This section was improved. However, I have some comments:

Provide a detailed description for tube formation assay and  migration assay. Describe specific parameters for image analysis. How was quantification performed?

Replication is mentioned inconsistently in all experiments. I don’t understand whether different rabbits were used for each pellet type or  were eyes treated independently?

For statistical analysis add information about software used, normality test, variance homogeneity, how sample size were determined.

Response: We sincerely thank the reviewer for the detailed and thoughtful comments. In response, we have added more detailed descriptions of the migration and tube formation assays, including the measurement and quantification methods. We also clarified the number of experimental replicates, provided a more specific explanation of the group allocation in the rabbit experiments, and elaborated on the statistical analysis methods.

Discussion

Rewrite discussions section. It has improved. However, I do have comments.

The discussion repeats information that already was explained in the introduction section. Interpret your actual findings. 

You mainly describe what you observed without deeper explanation. Please, provide comparative analysis of your results. 

Provide a critical analysis of limitations: small sample size (n=8 rabbits), lack of long-term angiogenesis data. No discussion. This weakens the results. Discuss clinical application and future directions. Discuss how cprp would be delivered into practice( injection or other?) .

What is the next step towards clinical translation?

Provide a critical comparative analysis of you results with previous studies.

Response: We sincerely thank the reviewer for the constructive feedback. We have thoroughly revised the Discussion section to address all the points raised.

Figures 

Still lack of consistent labeling, especially graphs

Add scale bars into migration and tube formation assays and microscopical images.

There is too much white space in some graphs. 

Check consistency of placing one or double asterisks on graphs.

Response: In addition, we have made all the necessary corrections to the figures as suggested. Thank you again for your valuable comments.

Reviewer 3 Report

Comments and Suggestions for Authors

I would like to thank authors for the effort they made to improve their manuscript. There are still some issues that are need to be improved:

Figure 1: Please check the graph once again. There should be no significant difference between the 10% cPRP and control group at 12 h. In the previous version also no difference was detected.

The legend of Figure 1 should be optimized cause it is not fully correct. As you have both * and ** significance levels, the standard way to report it is as below:

* p < 0.05, ** p < 0.01

Figure 2A. As I mentioned in the first review round, the title of each column is better shown on the top row rather than below.

Figure 2 legend: ** does not mean p<0.05, it means p<0.01. And I am not sure if the difference between the test groups in this graph and the previous group is less than 0.01.

Reference: Das D, Das T. The "P"-Value: The Primary Alphabet of Research Revisited. Int J Prev Med. 2023 Apr 26;14:41. doi: 10.4103/ijpvm.ijpvm_200_22. PMID: 37351025; PMCID: PMC10284198.

Fig 3 A and B: The same as previous images; move the title of each column on the top, correct the legend for P values of **. And check whether the difference between the test groups is really less than 0.01, which is mentioned as **.

Figure 4: I don’t know how you quantified the degree of angiogenesis in different groups, cause in my eyes there is no difference between days 7, 10, and 14 in the test groups! Apart from that, the pellet in control and 10% cPRP is not definable, at least I cannot define it in these groups.

Author Response

Author’s reply for Reviewer 3.

We would like to express our sincere gratitude to the reviewer for the thoughtful and thorough feedback. Please find our detailed responses below. All changes made based on the reviewer’s suggestions have been marked in blue in the revised manuscript.

Reviewer 3

Comments and Suggestions for Authors

I would like to thank authors for the effort they made to improve their manuscript. There are still some issues that are need to be improved:

Comments: Figure 1: Please check the graph once again. There should be no significant difference between the 10% cPRP and control group at 12 h. In the previous version also no difference was detected. The legend of Figure 1 should be optimized cause it is not fully correct. As you have both * and ** significance levels, the standard way to report it is as below:

* p < 0.05, ** p < 0.01

Response: We sincerely thank Reviewer 2 for the accurate and constructive comments. In accordance with the reviewer’s suggestion, we have revised the graph in Figure 1. Statistical significance is now indicated as * for p < 0.05 and ** for p < 0.01.

Comments: Figure 2A. As I mentioned in the first review round, the title of each column is better shown on the top row rather than below.

Figure 2 legend: ** does not mean p<0.05, it means p<0.01. And I am not sure if the difference between the test groups in this graph and the previous group is less than 0.01.

Response:  In response to the reviewer’s suggestion, we have repositioned the column titles to the top in Figures 2 and 3. Statistical significance is now indicated as * for p < 0.05 and ** for p < 0.01.

Fig 3 A and B: The same as previous images; move the title of each column on the top, correct the legend for P values of **. And check whether the difference between the test groups is really less than 0.01, which is mentioned as **.

Response:  In response to the reviewer’s suggestion, we have repositioned the column titles to the top in Figures 2 and 3. Statistical significance is now indicated as * for p < 0.05 and ** for p < 0.01.

Figure 4: I don’t know how you quantified the degree of angiogenesis in different groups, cause in my eyes there is no difference between days 7, 10, and 14 in the test groups! Apart from that, the pellet in control and 10% cPRP is not definable, at least I cannot define it in these groups.

Response:  We sincerely appreciate your insightful and detailed inquiry. The micropocket assay was conducted to confirm the pro-angiogenic effect observed in vitro within an avascular tissue, namely the cornea. As this was an exploratory study aiming to visually assess the angiogenic capacity of cPRP, quantitative analysis methods were not applied at this stage.

Regarding the reviewer’s comment on the visibility of the pellets, we would like to clarify that pellets without cPRP appear extremely clear and become indistinguishable from the surrounding corneal tissue after insertion. For this reason, we marked the pellet insertion site with an asterisk (*). On the other hand, pellets containing higher concentrations of cPRP become increasingly opaque, making them more visible in the cornea. This difference in visibility can be observed in Figure 4, where the 40% cPRP pellet appears more clearly than those with lower concentrations. We appreciate the reviewer’s attention to this point.